# Utilization of Biomasses from Landscape Conservation Growths Dominated by Common Ragwort (*Jacobaea vulgaris* Gaertn.) for Biomethanization

**DOI:** 10.3390/plants11060813

**Published:** 2022-03-18

**Authors:** Jürgen Müller, Denny Wiedow, Mohammad Said Chmit, Till Beuerle

**Affiliations:** 1Faculty of Agricultural and Environmental Sciences, University of Rostock, Justus-von-Liebig-Weg 6, 18059 Rostock, Germany; denny.wiedow@uni-rostock.de; 2Institute of Pharmaceutical Biology, Technical University of Braunschweig, Mendelssohnstr. 1, 38106 Braunschweig, Germany; s.chmit@tu-bs.de (M.S.C.); t.beuerle@tu-bs.de (T.B.)

**Keywords:** pyrrolizidine alkaloids, *Jacobaea vulgaris*, *Senecio jacobaea*, fermentation characteristics, PA degradation, energetic conversion, wet co-digestion, biogas yield

## Abstract

The highly toxic species common ragwort (*Jacobaea vulgaris* Gaertn.) prefers to migrate into protected dry grassland biotopes and limits the use of the resulting biomass as animal feed. There is an urgent need for a safe alternative use of the contaminated biomass apart from landfill disposal. We investigated the optional utilization of biomethanization of fresh and ensiled common ragwort biomasses and evaluated their energetic potentials by estimation models based on biochemical characteristics and by standardized batch experiments. The fresh and ensiled substrates yielded 174 L_N_∙kg^−1^ oDM methane and 185 L_N_∙kg^−1^ oDM, respectively. Ensiling reduced the toxic pyrrolizidine alkaloid content by 76.6%; a subsequent wet fermentation for an additional reduction is recommended. In comparison with other biomasses from landscape cultivation, ragwort biomass can be ensiled readily but has a limited energy potential if harvested at its peak flowering stage. Considering these properties and limitations, the energetic utilization is a promising option for a sustainable handling of *Senecio*-contaminated biomasses in landscape conservation practice and represents a safe alternative for reducing pyrrolizidine alkaloid entry into the agri-food sector.

## 1. Introduction

Although an autochthonous species in Europe [1], the increased spread of common ragwort (*Jacobaea vulgaris* Gaertn., syn. *Senecio jacobaea* L., Asteraceae) into the landscapes of northern Germany since the 1990s has invasive features [2]. The probable causes of this phenomena, namely climate change [3]; the increased spread of diasporas due to human activities [4]; and the establishment of diaspora banks [5], especially on fallow land [6], cannot be eliminated in the short term. Common ragwort (syn. tansy ragwort) is ubiquitous in ruderal habitats and on road verges [7], where it acts as a pollen supplier [8], provides host for many invertebrates [9], and fulfills landscape aesthetic functions via its flowering aspect. Despite these ecosystem functions, the species is also well known for its toxic 1,2-Dehydro-pyrrolizidine alkaloids (PAs), including their corresponding *N*-oxides (PANOs) [10,11,12]. PAs are widely known and well-studied liver toxins [13,14,15] that include a proven genotoxic potential [16] and are carcinogenic to mammals including humans [17,18]. The intrinsic deterrence effect of these secondary plant constituents against herbivores is problematic when the common ragwort migrates into farmland areas intended for livestock feeding [19,20]. Furthermore, beyond the apparent toxicologically relevant feed issue, high population densities of ragwort are today regarded as critical, since this represents an entry path for PAs into the food chain via co-harvesting of PA weeds together with food, medicinal herbs, or via nectar and pollen into bee-related foodstuffs [21,22,23,24].

Common ragwort prefers to migrate into extensively used grasslands with low nitrogen fertilization [25]. This constellation applies to the majority of nature conservation grasslands; therefore, these preservation areas suffer in particular from common ragwort mass infestations [26]. To prevent this noxious weed from outcompeting forage species and rare plants in such habitats, regulation measures are necessary [20] or even obligate [27]. Although effective [28,29], the use of herbicides is in general not an option for preservation areas; instead, twofold mowing is the recommended standard alternative measure [30]. According to Siegrist-Maag et al. [31] mowing should take place when half of the plants start anthesis. During this phenological stage, *J. vulgaris* has already accumulated a considerable aboveground biomass, including its toxic PAs/PANOs. In ragwort-dominant stands, up to 100 dt ha^−1^ may occur. Hence, removing the ragwort-infested biomass is necessary to avoid sward degradation and to enhance the biodiversity of the infested grasslands. However, the traditional use of conservation growths, such as forage for livestock, is often not possible due to its PA contamination [32,33,34]. Therefore, there is an urgent need for a sustainable and safe use of the contaminated biomass apart from disposing of it as landfill. In addition, as was recently demonstrated, careless landfill disposal may cause subsequent environmental contamination and distribution via leaching into surface waters [35].

As was recently shown, composting represents a disposal alternative, bearing the potential to safely degrade the PAs/PANOs [36,37]. However, this is a rather expensive alternative for larger quantities of biomass. Professional composting plants are rare in rural areas, and high standards for the quality and continuity of the feedstock are required [38]. These needs are rarely met by the naturally discontinuously accumulating amounts of conservation growths. In contrast to composting plants, biogas plants exist in Europe in high spatial density [39], and the produced “green” energy has a secure market acceptance. Hence, the utilization of ragwort biomass for biogas production seems to be a possible alternative. Concerning energy recovery, according to van Meerbeek et al. [40], biomethanization is the preferred option for summer growth of herbaceous plants. However, the exploitation of this bioenergy potential cannot always be realized right away using the fresh biomass. In order to process large quantities of short-term-available biomass, these landscape conservation growths are frequently ensiled before use [41]. Although there have been some ensiling experiments involving ragwort infested grassland biomass, those studies were mainly aimed at determining the rate of alkaloid degradation [32,42,43]. The results of these degradation studies did not show a consistent pattern or generally valid endpoints, and a large number of possible factors are discussed [32,33]. Thus far, the question of whether a low-loss conservation of common ragwort as a biogas substrate is possible has not yet been addressed. In addition, there are no studies on the economic efficiency of the methane yields or on the question of whether the fresh-cut substrate or the ensiled substrate is more favorable for biomethanization. Furthermore, it is very likely that the ontogenetic development stage of ragwort may influence the biogas yield, as is known from other herbaceous plants [44]. Hence, relevant for a practical implementation, the question of the most favorable cutting time is of great importance, not only for regulating ragwort, but also for biomethanization.

Therefore, the purpose of this study is to examine the potential use/safe disposal of common ragwort-dominated biomass for methane production in different states/conditions. 

In particular, we focus on the following key points:

(I)What are the substrate characteristics of a biomass from common ragwort at the recommended stage of cutting?(II)Can substrates dominated by common ragwort readily be ensiled without the addition of a carbohydrate-rich biomass?(III)To what extent does ensiling contribute to the reduction of pyrrolizidine alkaloids?(IV)Which specific methane yield can be attained by the common wet fermentation technique?(V)Which stage of ragwort development is best for controlling spreading of the plant, yet maintains the best option for efficient biomethanization?

These questions were addressed on the basis of two separate plant material collections. The first ragwort biomass material was used to investigate the fermentation process of lab-scale ensiling and biomethanization. The second collection was a time-series, conducted to study the influence of the phenological developmental stage of ragwort on its biochemical composition, which was necessary to estimate the stage-dependent bioenergy potential. 

## 2. Results

### 2.1. Characterization of the PA Pattern

The pattern and relative abundance of individual PAs of the *J. vulgaris* plant material were analyzed by GC-MS. Not surprisingly, all relevant PAs were 1,2-Dehydro-PAs of the senecionine-type (1,2-unsaturated, retroncine-, cyclic diester-, 12-membered ring-type) [11]. Figure 1 shows that the local chemotype of *J. vulgaris* was strongly dominated by the occurrence of erucifoline (79.5%), while the remaining 20.5% was distributed over five minor PAs ranging from 2.4 to 8.3%.

### 2.2. Further Biochemical Properties of the Ragwort’s Biomass

#### 2.2.1. Collection I—Fermentation

The mean crude ash (CA) content of the feedstock collected for fermentation purposes was relatively low (51.09 g kg^−1^ DM, sd = 0.52). The average development stage of early flowering was reflected by a high crude fiber (CF) content of 284.03 (sd = 10.77) g kg^−1^ of dry matter (DM). The sum of the cell wall components (neutral detergent fiber, NDF) was 457.93 (sd = 18.83) g kg^−1^ DM, of which the acid detergent fiber (ADF) accounted for 366.98 (sd = 14.10) g kg^−1^ DM. The average amount of 404.75 (sd = 19.78) g kg^−1^ DM of the organic matter was enzyme-insoluble (EISOM). Further biochemical properties relevant to ensiling success are listed in Table 1.

The crude protein (CP) fraction is responsible for the bulk of the buffering substances that counteract the decrease in the pH value during ensiling. Therefore, the moderate CP content of 10.87% DM did not indicate a large substrate-inherent resistance to lactic acid-initiated acidification. At the beginning of flowering, the substrate showed sufficient water-soluble carbohydrates (WSCH, 10.2% DM), which are the food sources of the lactobacilli. The WSCH:CP-ratio combines both characteristics into a single parameter that indicates moderate ensilability; i.e., sufficient sugar supply for the lactobacilli is countered by a notable buffering capacity in the substrate. Nevertheless, the pH value of 4.5 showed that under the laboratory conditions and at a high substrate moisture of 77.8% FM, the desired vigorous microbial conversion was readily achieved. The relatively high proportion of acetic acid, as well as the occurrence of ethanol and butyric acid, indicate a rather heterogeneous and non-homofermentative lactic acid fermentation. The ammonia content of 0.34 g kg^−1^ FM corresponds to a mean NH_3_-N content of 11.29 (sd = 7.62)%, both parameters indicates significant protein decomposition during ensiling.

#### 2.2.2. Collection II—Growth Stages

Collection II includes biomass stocks from common ragwort at the typical growth stages that match the cutting time schemes of *J. vulgaris*-infested grasslands. The ontogenetic development stages, sampling dates, matching grassland use options, and two metric traits characterizing the average plant development at the sampling dates are given in Table 2. The growth stage significantly influenced the average plant height and the dry matter content. The biochemical characteristics of these four stages are presented in Table 3. The ontogenetic stage affected all biochemical characteristics significantly. The direction of characteristic increase or decrease aligned with increasing plant development and was always consistent during the sampling period from BBCH 44 to BBCH 66.

While the fiber-related characteristics (CF, NDF, ADF, HC) increased with plant growth, a decrease in the energy-related CP, CL, and WSCH parameters was observed. In accordance, the crude ash decreased as a result of C-dilution via the increase in the plant biomass. It is noteworthy that within the flowering period, there was a notable increase in the proportion of cellolytically insoluble substance (EISOM).

### 2.3. PA Degradation during Ensiling Process

#### 2.3.1. Results of the Ensiling Experiment

A significant degradation of the sum of the total PA content (sum of PA/PANOs) after 92 days of lab-scale ensiling was detected (Figure 2). The extent of PA degradation was a remarkable 76.6% on average for the three laboratory silages. Beginning with a mean of 7962 mg in the starting material, an average of 1860 mg was still found per kg of dry matter in the silage after fermentation.

#### 2.3.2. Extent of PA-Degradation in the Context of Similar Studies

To evaluate this result, we compared our data to similar experiments where the PA degradation of a biomass containing species of the genus Senecio during ensiling were studied. The studies of Berendonk and Hünting [45], Candrian et al. [46], Gottschalk et al. [32], and Klevenhusen et al. [47] fulfilled the similarity requirements (see chapter 4.6 for details) and were used for the layout in Figure 3. The data used for this ordination are additionally listed in Appendix A
Appendix A.

With the exception of the PA-type senkirkine-dominated substrates, a gradient of decreasing degradation rates with decreasing moisture contents is apparent (Figure 3). Furthermore, there seems to be a trend towards an increase in the degradation rate with an increasing proportion of Senecio in the feedstock.

### 2.4. Methane Yields

#### 2.4.1. Results of the Batch Wet Co-Fermentation Test

The specific methane yields (SMY) of the fresh and ensiled biomasses from *J. vulgaris* were 173.6 L_N_∙kg^−1^ oDM and 185.1 L_N_∙kg^−1^ oDM, respectively. They were not significantly different (*p* > 0.05), despite a tendency of slightly higher SMY-values combined with lower SMY-variations of the ensiled feedstock (Figure 4). The distance between the mean values (top of the bars) and estimates of the methane potential based on the biochemical composition of the substrates (dots) is considerable for both ragwort substrates. In order to show a comparison to the most common gramineous-accompanying vegetation associated with J. vulgaris, the results of Meserszmit [48] have been integrated in Figure 4. These are *Arrhenatherum elatius* and *Festuca rubra*, the dominant grasses of the *Arrhenatheretum elatioris* grassland association, which is especially prone to common ragwort invasion. The feedstock of the companion grasses match that of the Senecio substrates, not only in terms of the plant sociological context, but also in terms of the time of harvest (July). In addition, the authors followed the same guidelines for the experimental determination of specific methane yields (VDI 4630).

The analysis of daily methane production rates provides insight into how methane yields are generated. Figure 5 compares the daily methane formation rates of the two different ragwort feedstock conditions, fresh (*J. vulgaris* fresh) and ensiled (*J. vulgaris* ensiled), throughout the whole experimental period of 30 days. Both variants reached the peak of daily methane formation, approx. 5 L_N_∙kg^−1^ oDM, at the same time, that is after 11 days. The dynamics of CH_4_-formation up to this point, however, are quite different: the fresh material started with a progressive formation rate, while the silage required a significantly extended start-up time. Conversely, methane production did not collapse quite as strong in the silage feedstock with an increasing lack of fermentable substance in the downstream phase. 

When looking at Figure 6, it becomes obvious that the methane yields of the initial phase were mainly due to high gas quantities at still low methane concentrations. With overall comparable carbon dioxide concentrations between the two feedstock conditions, the methane concentrations of the fresh substrate seem to be superior to those of the silage. From the fifth to the tenth day after starting the batch test, a stagnation of the two measured gas concentrations in the silage treatment was observed, which suggests a stronger presence of other non-measured gases (such as O_2_, NH_3_, NO_x_, H_2_S, i.a.). 

#### 2.4.2. Potential Methane Yields at Different Growth Stages

The potential methane yields (PMY) of Collection II (see Table 2) are shown in Figure 7. There was a significant effect of the developmental stage on PMY (*p* = 0.015 *). It is obvious that the potential of the substrate for methane production decreases with increasing phenological development. The difference between the cut for silage harvest and the late landscape conservation cut was considerable (106 L_N_∙kg^−1^ oDM, corresponding to 31%). Even the lowest PMY at the late developmental stage of BBCH 66 was higher than the measured specific methane yield in the batch test.

## 3. Discussion

### 3.1. Characteristics of Common Ragwort Biomass as Feedstock for Ensiling and Biomethanization

Apart from the discussion about the inhibiting effect of plants containing toxins in the biomethanization process [49], the ingredients of the substrate determine its utilization efficiency [50]. The knowledge of the composition of the lignocellulosic feedstock can help in choosing appropriate technical measures to achieve better hydrolysis which, in turn, would result in higher biogas yield and conversion efficiency [51]. Lignification of the substrate plant inhibits hydrolytic fermentation by hampering the microbes from attacking the cell wall carbohydrates. Therefore, many authors base their assessment of the biogas potential of unknown plant substrates on the lignin content as a key characteristic [52,53]. Nevertheless, choosing lignin content as the determining characteristic for biogas yield prediction is error-prone, as not only the amount of lignin, but also the incrustation of the polymeric lignin (crystallization), is relevant for microbial degradability [52,54]. In addition, the incorporation of the lignin into the plant tissue matrix and the binding via phenolic acids are of relevance [55]. Therefore, we decided to use enzyme insolubility (EISOM) as a key feature for estimating biogas potential, instead of lignin, as suggested by Weißbach [56]. 

The determined EISOM contents of *J. vulgaris* varied more (ranging from 132 to 534 g kg^−1^ DM) than those observed for the grass-dominated grassland biomass under comparable environmental conditions (range: 180–488 g kg^−1^ DM [57]). The enzyme-insoluble organic matter of the ragwort feedstock correlated with the phenological stage of development. Hence, the usefulness of *J. vulgaris* as a feedstock for bioenergy via the methanization pathway is strongly dependent on the average development stage of ragwort at the time of harvest. To our knowledge, there is no information on the enzyme insolubility of the organic matter (EISOM) of the species *J. vulgaris*. Only Frost et al. [58] and Purcell et al. [59] have provided information on the cell wall components of *J. vulgaris* depending on the developmental stage. According to Frost et al. [58], NDF-content increased from 37.6% DM at the rosette stage to 49.2% DM at flowering. Purcell et al. [59] did not specify developmental stages, but approximations can be drawn from the collection times given. Accordingly, plant material collected in early spring (in Ireland) had a cell wall content (NDF) of only 19.4% DM, whereas, at the end of August, the NDF content increased to 44.4% DM. The corresponding ADF contents in this study were 15.5% DM for the early collection and 34.8% DM for the late summer collection. ADF includes the fractions of lignin and cellulose and thus represents the recalcitrant part of a biomass for the microbiome involved in methanogenesis. The ADF contents in our study varied between 22.4 and 44.8% DM, representing a higher level. Those deviations could be explained by the fact that the spring sampling of the Irish survey took place very early in the season. Alternatively, the late summer sampling was probably the regrowth of an already pre-harvested area. Both conditions would lead to a biomass that is dominated by leaves and depleted in stems, which in turn would lead to lower levels of structural substances, such as NDF and ADF. In this light, these data are quite consistent with our results, confirming the overriding influence of the ontogenetic developmental stage on ragwort biomass composition, even across different environments. 

It is known from other herbaceous plants that their CP content responds more strongly to the N supply of the growing site [60]. Nevertheless, in the case of *J. vulgaris*, similar crude protein contents have been found across North America and Europe at comparable developmental stages. Debessai [61] reported a CP level of 13.1% DM in a not ontogenetically classified *J. vulgaris* collection, which was apparently collected at early flowering stages in Oregon (U.S.). Frost et al. [58] recorded higher CP contents (15.5% DM) in the rosette stage at sites in Idaho (U.S.), which corresponds well with the 15.4% CP that Purcell et al. [59] found in Ireland, also at an early developmental stage. The CP contents in our study fit very well with those of Frost et al. [58] and indicate a decrease in potentially buffering substances with the increasing maturity of the plants. 

Surprisingly, we observed the highest contents of water-soluble carbohydrates in the stage of vegetative development in Collection II. This is in contrast to Purcell et al. [59], who reported quite low WSCH contents in the spring-harvested ragwort (52 g per kg DM) and a considerably high level of 144 g per kg DM in the late summer collection. Both factors, different morphology (leaf/stem ratio and growth duration), and weather influences (solar radiation before collection) are possible explanations for these deviations. In any case, the WSCH content proves to be the most uncertain parameter for estimating the ensilability of ragwort biomasses. DM contents below 25–30% lead to the formation and release of silage effluent (McDonald et al. [62]). The DM content of the freshly harvested ragwort reached this threshold only at the end of flowering. Therefore, all earlier harvests need to be wilted before ensiling to avoid environmental threats, like the PA-leach-out risk, as well as additional energy losses associated with silage effluent run offs. In our study, the jars used for fermentation did not allow such effluent losses. Hence, under these ideal lab conditions, only gaseous ensiling losses of 0.23% DM were observed. Thus, in our experiments, the substrate characteristics of the fresh substrate differed only slightly from those of the silage, leading to comparable potential methane yields (shown as dots in Figure 4). Under practical conditions, however, this might be quite different.

### 3.2. Degradation of Pyrrolizidine Alkaloids and Their N-Oxides

The idea of using the process of ensiling as a means of detoxification of biomasses containing PAs was first conceived by forage scientists. Debessai [61] reported a reduction in PA content from an original 2454 to 584 µg kg^−1^ (−76%) after a 30-day experimental storage of ragwort-contaminated feedstock with the addition of urea preparations. In this study, grinding of the plant material was beneficial in reducing final PA contents as well. However, the alkaline environment achieved by urea ammonization, does not correspond to a lactic acid fermentation ensilaging process, which leads to an acidic environment. Therefore, we only compared our results with studies that also investigated lactic acid fermentation-based ensiling with regard to its effects on PA degradation. In their studies involving *Senecio alpinus*, Candrian et al. [46] found a decline in PA degradation from 92 to 54% with a decrease in the proportion of *Senecio* biomass in the silage substrate ranging from 100 to 3.5%, respectively. In the discussion of this finding, it was disregarded that the replacement of *Senecio* by a drier grass component was accompanied by an increase in the dry matter content while factor grading took place. Since the dry matter content significantly influences the overall microbial activity in silage fermentation processes [63], it may be that the observed decrease in PA degradation was also an effect of the overall reduction in microbial conversion activity too. Our results of the percentage PA degradation fit well with those of other studies if they are placed in an orientation framework (Figure 3) that takes both aspects, namely the ragwort percentage and the substrate moisture, into account. A number of authors who have experimentally studied the fermentative detoxification of PA-containing substrates, emphasize the importance of the specific PA type in the discussion of their results [32,43,47]. With the exception of the apparently recalcitrant otonecine-type PA senkirkine (a major PA in *Senecio vernalis* [47]), no striking differences in the degradation behavior of the general dominating senecionine-type PAs, such as erucifoline, senecionine, or seneciphylline, can be identified under comparable experimental conditions. This is not surprising since all these PAs share a similar basic chemical senecionine-type structure (1,2-unsaturated, retroncine-, cyclic diester-, 12-membered ring-type). In contrast to most other studies, our detection method was based on the detection of the core base structure of PAs and comprised all 1,2-unsaturated retronecine and heliotridine-type ester PAs [64]. This means that it would also cover for unknown toxic PA-metabolites that could be formed during the degradation process but still bear the general toxic principle of 1,2-unsaturated ester PAs.

Klevenhusen et al. [47] observed that the addition of homofermentative lactic acid bacteria did not increase the PA degradation rate but that the addition of molasses did. The addition of homofermentative lactic acid-producing inoculants results in a faster pH-drop, which strongly limits further microbial conversions, thereby shortening the fermentation-intensive phase of ensiling. Presumably, ensiling conditions that promote heterofermentative processes at the beginning, and allow lactic acid formation to stabilize the ensiled material by lowering the pH value a little later, are more conducive to PA degradation. Further ensiling experiments would also be desirable in this area, e.g., with staggered buffering of otherwise identical PA-containing and PA-type classified substrates.

### 3.3. Specific Methane Yields of Ragwort Biomasses

In a previous study, we were able to show that the combined process of ensiling/biomethanization leads to a significant and reliable reduction of PAs/PANOs by 91–99% of the starting material [37]. These findings are only useful for practical implementation if the *S. vulgaris* biomass can be converted into green bioenergy with economic success. This, in turn, requires solid knowledge of the substrate-specific methane formation potential, as well as of the behavior in the biogas process [52]. We found specific methane yields of only 173.6 L_N_∙kg^−1^ oDM (fresh) and 185.1 L_N_∙kg^−1^ oDM (silage) in the standard batch digestion test. These were significantly lower yields than can be achieved by herbaceous cultivated crops, such as catch crops (250–350 L_N_∙kg^−1^ oDM) [50]. Even grass-dominated and extensively used grassland growths have higher specific methane yields of 197–221 L_N_∙kg^−1^ oDM [48]. Consequently, the usability of ragwort-dominated growths harvested in the flowering stage has to be considered critical, even if the prime costs of the biomass production were not credited to the biogas generation. There was a greater discrepancy between the potential energy yield and the yield measured in the batch tests (Figure 4) than the usually reckoned 15%. This is an indication that the microbiome in the test did not work as expected and thus seems to confirm that PAs could disrupt the complex and sensitive microbiological conversion processes of the methanogenesis [49]. However, Chmit et al. [37] could identify neither a pronounced lag phase nor a correlation between PA levels and methane formation. Therefore, other ingredients, such as inhibitors or even an insufficiently adapted microbiome of the inoculum, must be responsible for the reduced methanogenesis. The inoculum we used originated from a biogas plant based on renewable raw materials. In order to prove or exclude the aspect of inhibition, the adaptability of the microorganisms would have to be examined in continuous tests over a longer period of time. In this way, the microorganisms could adapt to the substrate, and the accumulation of inhibitory substances could also be better detected. 

The development of the gas formation in the course of the batch test can provide information about the process dynamics. As shown in Figure 5, the methane formation kinetics of fresh material and ensiled material differed somewhat, despite a comparable basic pattern. The higher acidity of the silages, in combination with a lack of buffering substances, was probably responsible for the delayed increase in the gas formation rate in the first week [65]. In contrast, the degression in the gas formation after the consumption of the microbially more-easily-accessible carbohydrate sources was not as strong in the silage variants after almost two weeks, most likely due to a certain pre-digestion of the cell wall components by the enzymatic activity of the silage microflora [66]. This is also indicated through the somewhat lower EISOM values of the silages compared to the fresh substrates. The non-ensiled biomasses tended to have higher methane contents in the biogas, although, according to Herrmann et al. [67], the ethanol and butyric acid formed during ensiling should lead to higher methane contents in the ensiled substrates. However, both fermentation products could only be detected in limited quantities. 

The strong dependence of the potential methane yield on the developmental stage of the plants from Collection II is not surprising and is also reported for a number of other plant species [44,68,69]. While the energy yield per ha in the case of cultivated bioenergy crops plays a decisive role in the determination of the time of harvest [69], the compatibility of landscape conservation objectives with biomass usability has priority in the case of conservation growths. When ragwort-dominated areas are cut at the time of full flowering in order to suppress the noxious weed, the resulting biomass can hardly be used for methane production in an economically viable way. The challenge for the bioenergetic use of *J. vulgaris* is to schedule the time of harvest in such a way that the regulating effect for the containment of this invasive species is compatible with the requirements on substrate quality for efficient methane production. If this succeeds, an invasive burden can be turned into a bioenergetic opportunity. Already, at the recommended cutting time of early flowering [31] and with a growth height of about 57 cm, only marginal methane yields are achieved. Therefore, we recommend harvesting such growths somewhat earlier to ensure sustainable energy production with simultaneous detoxification at a development stage not earlier than BBCH 48, but not later than BBCH 56. Cuts taken too early do not provide enough biomass to justify the harvesting costs, are more difficult to ensile, and, according to Berendonk et al. [42], also possess lower PA degradation rates during ensiling. 

### 3.4. Synthesis 

We have investigated the usefulness and behavior of ragwort-dominated biomasses for the fermentation processes of ensiling and biomethanization based on a collection of a larger amount of uniform plant material (referred to as Collection I). This corresponded in its phenological development to the sensitive stage of full flowering, recommended by weed researchers, when cutting most severely affects the vitality of the invasive species. In addition, we have investigated the expected behavior of the ragwort feedstock at different development stages on the basis of a second time/development-based collection of plant material (referred to as Collection II). Evaluation of ensilability was done by means of biochemical characteristics and, in the case of biomethanization, with the help of a validated model approach based on enzymatic degradability. In the preceding sections of this discussion, the results of both collections and aspects were discussed in order to comply with the complex character of the matter. The final conclusions for the practical significance in landscape management and biomass utilization will be summarized in Section 5. 

Further research on the topic would be desirable. In particular, the measurement of methane yields in a continuous-flow-digester, instead of the batch tests, would generate additional useful information. 

## 4. Materials and Methods

### 4.1. Chemicals and Reagents

All chemicals used were purchased from Roth (Karlsruhe, Germany) and Sigma-Aldrich (Seelze, Germany) and were of HPLC-grade purity or redistilled before use. Lithium aluminum hydride solution (1 M) in THF and pyridine, both in AcroSeal quality, were acquired from ACROS Organics (Geel, Belgium). Strong cation-exchange solid-phase extraction cartridges (SCX-SPE) were obtained from Phenomenex (Aschaffenburg, Germany). Isotopically labeled internal standard (IST) 7-*O*-9-*O*-dibutyroyl-[9,9-^2^H_2_]-retronecine was synthesized in our laboratory [64].

### 4.2. Plant Material 

The plant material was harvested in two collections of already investigated J. vulgaris populations near Rostock (northeastern Germany). Collection I (fermentation collection) was collected on 5 July 2019 at the main developmental stage of BBCH 57 (beginning of the flowering period), according to Meier [70]. The fresh biomass was brought to the laboratory immediately and chopped with a commercial garden shredder. The shredded biomass was divided into three subsets, which then served as replicates. From each of these stocks about 500 g of fresh mass was drawn to determine the dry matter content and the biochemical analysis of the starting material. Subsets were taken from the bulk of the remaining plant material, both for biomethanization as fresh substrate (approx. 1500 g FM per sample, stored at −22 °C until the start of the batch trial) and as silage material (approx. 3000 g FM per sample, directly ensiled thereafter). 

Collection II (stage collection) was carried out during the 2021 growing season with the aim of collecting common ragwort at its typical physiological development stages that, in practice, correspond to the usual cutting times of *J. vulgaris* - infested grasslands. On the following dates, 1 June (silage cutting), 8 June (early hay cutting), 16 June (late hay cutting), and 5 July (landscape conservation cutting), approximately 2000 g of ragwort plants were collected, and their growth height, as well as their development stages, were determined. Subsequently, the plant material was dried at 55 °C to a constant weight and ground to 1 mm sieve width using a hammer mill (Brabender, Duisburg, Germany).

### 4.3. Silage Preparation

The chopped *J. vulgaris* biomass was ensiled in three replicates in 3 L lab-scale silos (jars). The jars were cleaned and sterilized (180 °C, 8 h) before the substrates were filled in by hand. Fill weights of 1552–1563 g FM were realized, corresponding to a mean storage density of 0.52 g cm^−3^ FM. The filled jars were closed air-tight with a rubber-lined lid that was fixed by clips and stored for 92 days at 16 °C in a dark room. This was designed to allow gas to be released under pressure but prevented ambient air from flowing in. The jars were weighed after 30 days and again after the end of the ensiling period to determine the ensiling loss gravimetrically. After 92 days, the ensiled substrates were sealed air-tight in plastic bags and stored at −22 °C before analyzing their biochemical composition, including fermentation profiles. 

### 4.4. Biochemical Analysis

#### 4.4.1. Substrate Characteristics 

DM content of the ragwort biomass was determined by oven drying at 55 °C to a constant weight, immediately after collection and additionally from the feedstock before and after ensiling. We analyzed the water-soluble carbohydrates (WSCH) and the enzyme-insoluble organic matter (EISOM) by near-infrared reflectance spectroscopy (NIRS, Bruker^®^ MPA, Bruker, Ettlingen, Germany) with the enzymatic method, according to de Boever et al. [71], as the reference for EISOM. The dry combustion technique (Elementar^®^ Analyzer Vario Max CNS, Elementar, Sprendlingen, Germany) has been used to determine crude protein contents (CP, N × 6.25). Neutral detergent fiber (NDF), acid detergent fiber (ADF), and crude fiber (CF) were determined by wet chemical analyses using a Fibretherm^®^, (Gerhardt, Königswinter, Germany). Hemicellulose contents have been estimated through the difference between NDF and ADF concentrations. Crude ash (CA) content was determined after incineration at 550 °C for 5 h in a muffle furnace. Volatile solids (VS) were calculated by the difference between total solids (TS) and the amount of CA.

#### 4.4.2. Silage Fermentation Characteristics

The fermentation characteristics of the lab silages were determined according to the VDLUFA guidelines [72]. After thawing the frozen silage samples at room temperature, silage extracts were prepared from 50 g silage and 200 mL deionized water. The pH values of these extracts were measured potentiometrically by a calibrated pH analyzer (WTW pH3210, Xylem Analytics, Weilheim, Germany). Acetic and butyric acids were quantitatively detected by gas chromatography (6890 Series GC-FID, Agilent Scientific Instruments, Santa Clara, CA, USA).

#### 4.4.3. Quantification of the Total PA/PANO Content 

Plant material was homogenized (dried, mixed, and milled). Each sample of 300 mg was soaked in a 15 mL polypropylene conical centrifuge tube using 11 mL H_2_SO_4_ 0.05 M and 40 µL of internal standard solution (2 µg/mL in methanol) and was added to each tube. The tubes were thoroughly mixed and extracted over night using a continuous tube rotator (Multi Bio RS-24, Biosan, Riga, Latvia) with the following settings: orbital: 21/01, reciprocal: 15/01, vibrio: 5/1, duration: 14 h. Sample preparation for analysis of the total PA content of the products was determined by using a modification of the method of Cramer et al. [64] and Letsyo et al. [73]. During the course of sample preparation all 1,2-unsaturated retronecine- and heliotridine-type ester PAs/PANOs were converted into the corresponding core structures, i.e., retronecine and/or heliotridine. After derivatization, these analytes were analyzed by HPLC-ESI-MS/MS, generating a single signal that could be quantified by the use of the added isotopically labeled IST, allowing calculation of the total sum of the PA/PANO content of the sample (Cramer et al., 2013). Data analysis and integration was achieved with Analyst 1.6.2 Software (Applied Biosystems MDS Sciex, Darmstadt, Germany).

#### 4.4.4. Sample Preparation for GC-MS Analysis 

Plant material of fresh *J. vulgaris* plant material was frozen (−20 °C), lyophilized, powdered, and homogenized with a coffee grinder. The homogenized powders were stored at room temperature in the dark until sample preparation and analysis. The powdered plant material was extracted with 0.05 M sulphuric acid (3 × 2.0 mL) after adding an appropriate volume of heliotrine as an internal standard solution (0.1 mg from MeOH stock 10 mg/mL stored at −20 °C). The extraction process was repeated three times and centrifuged (>3500× *g* for 5 min). The acid-aqueous supernatants were combined and reduced with zinc dust (~100 mg) for 3 h with continuous stirring at pH~2 to convert existing PANOs to the corresponding tertiary PAs. After 3 h, the mixture was adjusted to alkaline (pH~10) with ammonia solution (25%) and then eluted from 3 to 6 g Isolute^®^ column (1 mL aqueous solution/g Isolute) with 25 mL dichloromethane. The organic phase, after the Isolute clean-up process, was dried at room temperature and reconstituted in exactly 1.0 mL of methanol for GC-MS analysis.

#### 4.4.5. GC-EI-MS PA Pattern Analysis of Raw Plant Materials

An Agilent Technologies 6890 N (Agilent Technologies Deutschland GmbH, Böblingen, Germany) gas chromatograph was coupled to an Agilent Technologies 5975B mass spectrometer. Separation of the analytes was performed on a ZB-1MS capillary column (30 m × 0.25 mm; ft 0.25 µm; Phenomenex, Aschaffenburg, Germany). The following temperature program was used: 100 °C (3 min)–6 °C/min–310 °C (3 min). The following MS settings were applied: ionization voltage of 70 eV, ion source temperature of 230 °C, and interface temperature of 290 °C. A set of co-injected hydrocarbons (evenly numbered C14–C32) was used to calculate the retention indices by linear extrapolation, as described by Frölich et al. [74]. The PA compounds were identified by retention indices and an in-house mass spectra library of authentic reference compounds. PA patterns of the individual plant samples were compared using the Agilent ChemStation software (Version D.03.00.611). The respective ratio (relative abundance, RA) of each PA was calculated based on its TIC-area signal in relation to the sum of all TIC-area signals of all PAs of the sample.

### 4.5. Biogas Yield Determination

#### 4.5.1. Potential Methane Yield Estimation 

The potential for methane formation was calculated according to Weißbach [56] using the following estimation equations based on two main biochemical parameters of the substrates:VS = 1000 − (CA) − 0.62 (EULOS) − 0.000221 (EULOS)^2^(1)

The substrate’s amount of fermentable organic substances (VS g kg^−1^ DM) was estimated using Equation (1). The substrate-specific biogas (BGY) and methane (CH_4_Y) yield potentials of the ragwort biomass as feedstock substrates were derived from vs. according to Equations (2) and (3), accordingly.
BGY = 0.80 (VS)(2)
CH_4_Y = 0.42 (VS)(3)
where BGY and CH4Y are given in norm liter per kg (NI kg DM^−1^) and are corrected for volatile fatty acids (VFA) [75].

#### 4.5.2. Batch Wet Co-Fermentation Test 

A batch test according to VDI 4630 [76] was carried out to determine the biogas and methane potentials. For this purpose, digesters of 1 L batch were used, which were placed in a water bath at 38 °C. Each sample was tested in triplicate. The ratio of organic matter of the sample to organic matter of inoculum was <0.5 and was used to calculate the initial weight. After the addition of the substrate inoculums mixture, the digesters were closed and flushed with N_2_ atmosphere. The produced biogas from each replicate was stored in a gas bag. The methane and carbon dioxide contents of the biogas was analyzed every three days with a biogas analyzer (bm 2000, Ansyco^®^, Karlsruhe, Germany). The volume of biogas was measured via gas meter (Ritter Apparatebau GmbH, Bochum, Germany). The biogas and methane potentials were corrected as standard norm liters (273.15 k and 1013.25 mbar) and reported for each kilogram fresh matter or organic matter. The test was ended until the daily amount of biogas produced corresponded to 1% of the total amount of gas (here, after 30 days).

### 4.6. Data Analysis

Biochemical characteristics are presented as means and corresponding standard deviations of the mean (sd) in brackets. The test characteristics affected by the factors “substrate condition” (Collection I) and “development stage” (Collection II) were first tested for normal distribution using the Shapiro–Wilk test. For a given normal distribution, one-way analysis of variance (ANOVA) followed by an F-test was applied. When the trait values were not normally distributed and neither log nor sqrt transformations achieved a normal distribution, the Kruskal–Wallis test was preferred. In the case of significant factor effects (*p* < 0.05), post-hoc mean comparisons among the factor levels were calculated. After an ANOVA, the HSD Test was used for this purpose. After the non-parametric alternative Kruskal–Wallis test, the Dunn test was applied. The α-level was set to 0.05. All statistical analyses were performed by scripts using the R environment version 3.6.3. [77].

Criteria for the comparative analysis to place our own results on PA degradation by ensiling into the context of current knowledge (Figure 3) were as follows: (1) laboratory ensiling conditions similar to our study, (2) information on the proportions of PA-containing species in the substrate, (3) identification of the dominant biochemical PA type, (4) an ensiling period of 90–105 days, and (5) precise information on the dry matter content of the ensiled material.

## 5. Conclusions

The substrate of the biomass from common ragwort is characterized by a strong dependence of the utilization-determining composition in relation to the ontogenetic development stage. In the course of the phenological development of common ragwort until the end of flowering, younger side shoots and flower sets are obviously not able to compensate for the progressive lignification of the structural substances. At the recommended stage of cutting, the proportions of structural substances have reached such an extent that their biochemical conversion in wet fermentation processes is limited.

Moderate levels of crude protein and sufficiently high levels of water-soluble carbohydrates for lactic fermentation allow for ensiling of common ragwort-dominated feedstock without the addition of WSCH-rich substrates, even in the stage of full flowering. However, the level of ensiling technology has to be advanced, especially in terms of rapid silo filling, careful feedstock compaction, and prompt covering in order to ensure early onset of anaerobic conditions. The low dry matter content of a freshly harvested ragwort biomass is always below the threshold value of 28% dry matter. Hence, direct ensiling is not recommended. The harvested plant material must be wilted before chopping to avoid effluent runoffs from the feedstock.

For the senecionine-type PAs, ensiling proved to be a suitable method of PA degradation. The prerequisite for this is a rather low dry matter content. Hence, excessive high wilting degrees need to be avoided. As a result, the target corridor for significant PA reduction via ensiling is quite narrow (28–32% DM). Since ensiling alone cannot guarantee PA degradation to a level that reliably eliminates contamination risks, subsequent biomethanization is an advisable option. After biogas processing, the PA-containing feedstock is decontaminated to such an extent that the digestate can be applied as organic fertilizer to grain crops without further risks to the agri-food chain. This concerns not only the PA contamination risk, but also the risk of ragwort seed dispersal with the digestate, which is a very efficient method for preventing germination.

Specific methane yields of only 173.6 LN∙kg^−1^ oDM (fresh) and 185.1 LN∙kg^−1^ oDM (silage), as they occur at the time of full flowering (at this point, the recommended cutting scheme), are in the lower scale level of landscape conservation growth and thus in the border area of economic usability. Ragwort substrate utilization can be lucrative for a biogas plant operator under the condition of short transport distances from the feedstock to the plant and exemption from the prime costs of the biomass production. By factoring in the safe treatment and degradation of PA-contaminated biomasses of an otherwise critical plant material, this usage option is definitely advisable. Our recommendation for a utilization-friendly application and control of common ragwort mass infestations areas is linked to the average growth heights of *J. vulgaris* from about 45 to 55 cm, largely irrespective of its state of flowering. However, if the proportion of ragwort does not dominate the stand, it is possible to cut at full flowering, if the accompanying gramineous vegetation can compensate for the poorer microbial degradability of the fermentable content.

## Figures and Tables

**Figure 1 plants-11-00813-f001:**
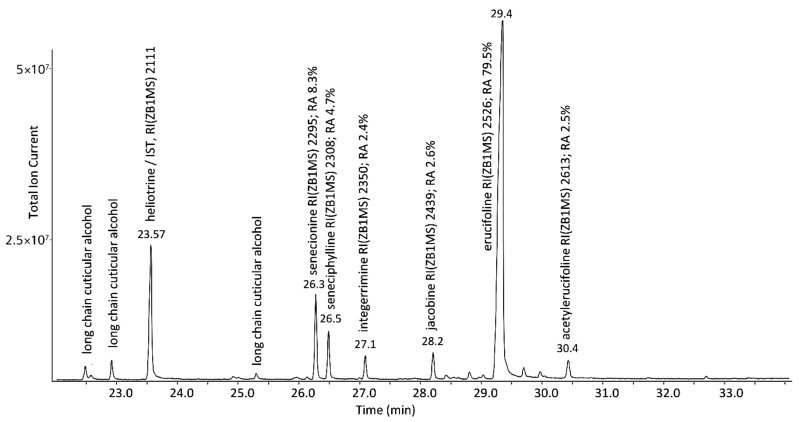
Total ion chromatogram (TIC) of a GC-MS analysis for identification and determination of the relative abundance (RA) of the PA-bouquet of *J. vulgaris* plant material utilized in this study.

**Figure 2 plants-11-00813-f002:**
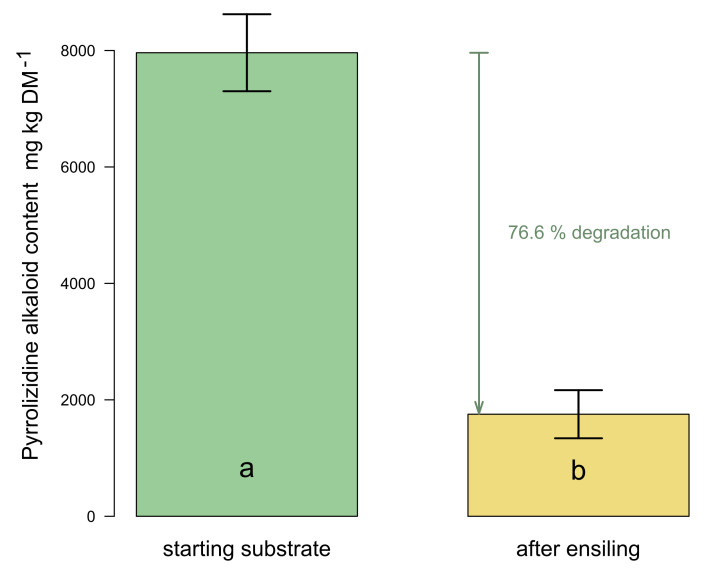
Extent of degradation of the total amount of PAs/PANOs after 92 days of ensiling. Different letters indicate significant differences in PA/PANO content (Dunn-Test, *p* < 0.001). Error bars represent the standard deviations of the means.

**Figure 3 plants-11-00813-f003:**
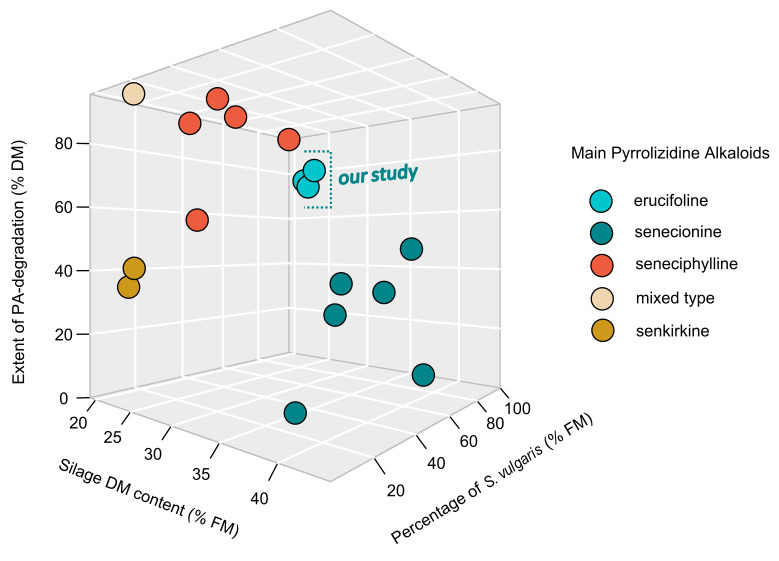
Extent of PA/PANO-degradation during ensiling of our study and comparable studies with PA-containing *Senecio* species. Results are ordinated according to the dry matter content of their silage substrates and the percentage of the *S. vulgaris* resp. *Senecio* biomass in the ensiled feedstock. The main types of the pyrrolizidine alkaloids are color coded (see legend for details).

**Figure 4 plants-11-00813-f004:**
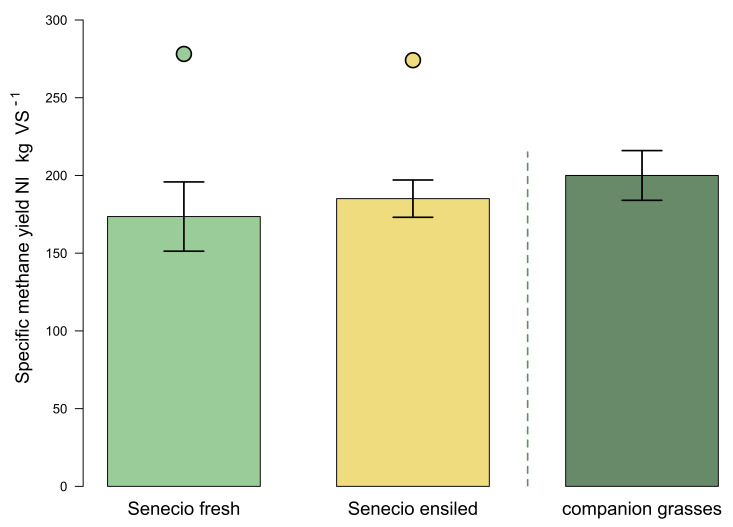
Specific methane yields of fresh and ensiled biomasses from *J. vulgaris*. Presented are the means and the corresponding standard deviations as error bars. Dots mark the amount of specific methane yield potential according to Weißbach (2008). For comparison, the yield of the prevalent companion grasses according to Meserszmit et al. (2019) is additionally shown.

**Figure 5 plants-11-00813-f005:**
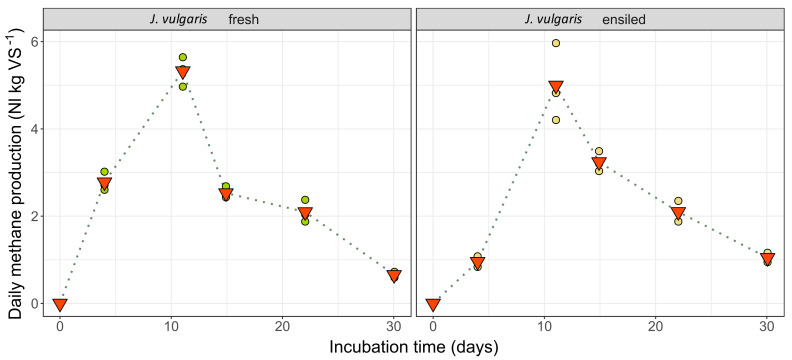
Daily methane production (expressed as norm liters per kg volatile matter) throughout the batch trial period of 30 days. Red triangles present the mean methane yield of the respective measurement event. Dots indicate the single results of the three batch vessels that served as replicates.

**Figure 6 plants-11-00813-f006:**
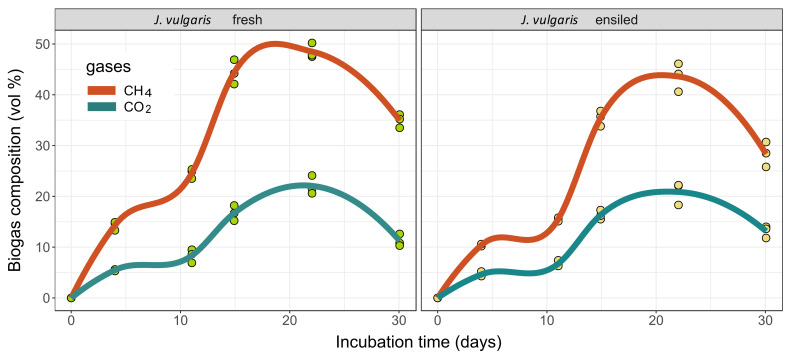
Volume percentages of the two main biogas components methane (CH_4_) and carbon dioxide (CO_2_) during the course of the batch wet fermentation period. Dots indicate the measurement results of the individual batch vessels in triplicate. Trendlines were constructed using local regression approaches (loess).

**Figure 7 plants-11-00813-f007:**
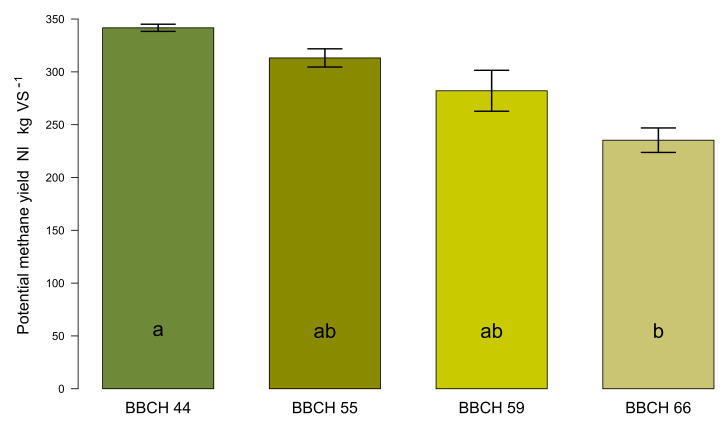
Potential specific methane yields (expressed as norm liters per kg volatile substrate) of common ragwort feedstock harvested at different developmental stages. Different letters inside the bars indicate statistically significant differences between the four stages ranging from BBCH 44 to BBCH 66. Error bars represent standard deviations of the mean.

**Table 1 plants-11-00813-t001:** Ensiling characteristics of biomass from landscape conservation growths dominated by common ragwort (*J. vulgaris*).

Status of Ragwort Biomass	Feedstock Characteristics	Means (sd)
Chopped starting material before ensiling	Parameters of Ensilability	*n* = 3
	Dry matter content at start (DMC, g kg^−1^)	224.3 (3.79)
	Crude protein (CP, g kg^−1^ DM)	108.7 (0.41)
	Water-soluble carbohydrates (WSCH, g kg^−1^ DM)	102.3 (11.72)
	WSCH:CP-ratio	0.94 (0.12)
Substrate after 92 days of lab-scale ensiling	Fermentation patterns	
	Dry matter content after ensiling (DMC, g kg^−1^)	222.5 (0.12)
	pH-value	4.50 (0.00)
	Lactic acid (g kg^−1^ DM)	34.40 (4.48)
	Acetic acid (g kg^−1^ DM)	8.65 (0.83)
	Ethanol (g kg^−1^ DM)	1.60 (0.01)
	Butyric acid (g kg^−1^ DM)	<0.50
	Ammonia (g kg^−1^ FM)	0.34 (0.01)

**Table 2 plants-11-00813-t002:** Ontogenetic characterization of the developmental stages of *J. vulgaris* at different cuttings during the season.

Stage of Development (BBCH) ^1^	Vegetative Parts Reached ca. 40% of Final Size (44)	First Flowers Visible, Largely Still Closed (55)	Inflorescence Fully Emerged (59)	Flowering Nearly Finished (66)	Effect of Stage ^3^
Sampling Date	1 June 2021	8 June 2021	16 June 2021	5 July 2021	
Corresponding grassland use	cut for silage	cut for hay	late cut for hay	conservation cut	
Averaged plant height (cm) ^2^	33.70 (3.91)	56.80 (0.43)	84.33 (4.41)	99.33 (3.26)	*p* < 0.001 ***
Dry matter content (g kg^−1^) ^2^	116.79 (3.70)	156.98 (0.88)	192.57 (7.34)	267.26 (6.88)	*p* < 0.001 ***

^1^ General BBCH-scale according to Meier (2001); ^2^ sample mean with standard deviation in brackets; ^3^ results of the one-way-ANOVA; ^***^ significant at *p*-values less than 0.001.

**Table 3 plants-11-00813-t003:** Constituent characteristics of common ragwort (*J. vulgaris*) biomass depending on its averaged phenological stage. The means are presented with their standard deviations in brackets.

Stage of Development (BBCH) ^1^	BBCH 44	BBCH 55	BBCH 59	BBCH 66	Effect of Stage ^2^
Crude ash (CA, g kg^−1^ DM)	100.73 (2.37)	76.13 (0.90)	55.60 (3.22)	45.30 (1.34)	*p* < 0.001 ***
Crude protein (CP, g kg^−1^ DM)	155.92 (4.32)	118.92 (6.20)	93.04 (3.17)	79.95 (4.48)	*p* < 0.001 ***
Crude fiber (CF, g kg^−1^ DM)	161.28 (7.02)	220.28 (10.11)	293.31 (14.93)	340.30 (7.78)	*p* < 0.001 ***
Crude lipid (CL, g kg^−1^ DM)	23.93 (1.35)	22.58 (2.15)	18.35 (1.02)	18.02 (0.53)	*p* = 0.002 **
Neutral detergent fiber (aNDF_OM_, g kg^−1^ DM)	284.39 (8.79)	388.12 (16.04)	479.97 (22.89)	537.45 (12.12)	*p* < 0.001 ***
Acid detergent fiber (ADF_OM_, g kg^−1^ DM)	224.26 (6.11)	299.65 (12.96)	378.60 (14.20)	428.86 (7.08)	*p* < 0.001 ***
Hemicellulose (HC, g kg^−1^ DM)	60.13 (6.26)	88.46 (4.15)	101.37 (8.69)	108.59 (5.06)	*p* < 0.001 ***
Water-soluble carbohydrates (WSCH, g kg^−1^ DM)	106.15 (3.34)	93.00 (8.15)	77.58 (1.71)	42.15 (4.71)	*p* < 0.001 ***
Enzyme-insoluble organic matter (EISOM, g kg^−1^ DM)	131.94 (8.63)	262.70 (14.22)	386.38 (32.71)	534.36 (14.54)	*p* < 0.001 ***

^1^ General BBCH-scale according to Meier (2001); ^2^ results of the F-test (one-way-ANOVA) or Kruskal–Wallis test—depending on data distribution; ^**^ significant at *p*-values less than 0.01; ^***^ significant at *p*-values less than 0.001.

## Data Availability

Data are included in this article. In the event of justified inquiries in the interest of science, the authors will provide the original data.

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
