# Peer review of "Utilization of Biomasses from Landscape Conservation Growths Dominated by Common Ragwort (*Jacobaea vulgaris* Gaertn.) for Biomethanization"

_plants, 2022, doi:10.3390/plants11060813_

Round 1
Reviewer 1 Report
There is a need for an alternative safe use of the contaminated biomass apart from landfill disposal for the highly toxic species common ragwort (Senecio jacobaea L.) The optional utilization of biomethanization of fresh and ensiled common ragwort biomass was investigated and its energetic potential was evaluated by estimation models based on biochemical characteristics and by standardized batch experiments. Ensiling reduced the toxic pyrrolizidine alkaloid content by 72.4 %, a subsequent wet fermentation for an additional reduction is recommended. In comparison with other biomasses from landscape cultivation, ragwort biomass can be ensiled readily, but has a limited energy potential if harvested at its peak flowering stage. The manuscript is interesting, well written and designed. The end of Introduction part is clearly pointing out the novelty and the specific topics that are addressed in the manuscript.
Minor remarks:
The results of GC-MS analysis (paragraph 4.3.4.) could be presented as % of identified compounds. Namely, Figure 1 is presenting TIC chromatogram of identified compounds, but the table with their percentages or relative amount determined should be presented. The title of Figure 1 should be changes since it represents total ion chromatogram (TIC), not GC-MS analysis (mass spectra are not presented).
How in Figure 3 the extend of pyrrolizidine alkaloids degradation was determined? Can the results be presented as table as well to present degradation of each alkaloid: erucifoline, senecionine, seneciphylline, mixed type and senkirkine? It would be interest to follow degradation of individual components as well, not only as their total content.
Author Response
The results of GC-MS analysis (paragraph 4.3.4.) could be presented as % of identified compounds. Namely, Figure 1 is presenting TIC chromatogram of identified compounds, but the table with their percentages or relative amount determined should be presented. The title of Figure 1 should be changes since it represents total ion chromatogram (TIC), not GC-MS analysis (mass spectra are not presented).
Response: The ratio (relatives amounts in %) for each PA is already given in the figure, right next to the peaks itself. Since there are only 6 PAs we think an extra table for this is not necessary and all the information can be accessed barrier free. The details were added to the Material and Method section as well. The legend of Fig. 1 was changed according to the reviewer’s recommendations referring to the TIC.
How in Figure 3 the extend of pyrrolizidine alkaloids degradation was determined? Can the results be presented as table as well to present degradation of each alkaloid: erucifoline, senecionine, seneciphylline, mixed type and senkirkine? It would be interest to follow degradation of individual components as well, not only as their total content.
Response: In our case, we applied a sum parameter approach to quantify the PAs/PANOs. This comes with the disadvantage that we lose the structural information of the original PAs/PANOs, however we gain the advantage that we comprehensively cover also for possible degradation PA/PANO-metabolites. From all that is known from literature, in our case, all PAs were members of just one type (senecionine-type), and no otonecine-type PAs (or other types) were present. Hence, we do not expect unusual degradation behavior and all PAs should be degraded at a similar rate. The method and the corresponding literature of the PA-degradation during ensiling is already included in the discussion section (3.2) and the relevant literature is cited.
Reviewer 2 Report
The manuscript is particularly strong regarding less studied topic and the experimental set up on ragwort biomass and methane production. The manuscript regarding the topic and results presented is of interest to plant scientific community and revisions based on the comments below are recommended before considering for publication.
Major comments
- I would be better to discuss/present the result in the order of the aims. That were discussed in line 94-104. In the current version, the flow is not the same as presented.
- In addition, the link between sub-sections in RD section is missing. Even though the sub-sections are nicely discussed, however, the are not sell connect with each other. For example, what we will lean from section I?, and what is the additional outcomes from section II? Then are section I and II connected? Or how do they complete each other. Thus, this kind linkage is necessary!
- Degradation rate and half-life either presented nor discussed, for example in section ‘’2.3.1. Results of the ensiling experiment’’
- Inter-specific changes / metabolic reduction of N-oxides for PAs are not disused.
- The unit / abbreviation is not mention before, consider define the abbreviation when mentioned for the first time…. Please check throughout the manuscript to define the abbreviations…….
- Line 94-104, the aim or hypothesis of the study is clear, however the approach is missing ….
- Lake of scientific literature for section 3.4. Synthesis
- PA concentration/content; is not clear the data discussed in result section is generated by HPLC or GC, this need t be clear through out the manuscript.
Specific comments
Table 1. You need to define any acronyms or term that used; Dry matter content at start, and Water-soluble carbohydrates with a footnote……and (sd) is from n=? measurements
Line 140: Did you define ‘’ WSCH:XP-ratio’’ before, or is it a typo error of WSCH:CP-ratio
And ‘’ moderate ensilability’’ is rather descriptive, can you define the moderate with maybe scientific figures.
Table 2. Define the numbers is brackets for BBCH, is it number of days? replicate?
For ‘’effect of the stage’’ compered with what? Its is not clear!
Line 154-158: how constituent characteristics correlated with the plant growth? Or for example do we know why the ‘’crude ass’’ decreased with growing season, the same for rest of the constituents
Table 3, define the units. Was the correlation for ‘’ effect of the stage’’ was positive for all constituents.
Line 169-12: check the grammar! And some other places…
Line 184-189: the sentences are hard to follow, please consider rewarding them.
Line 195-205: The approach has not been discussed in MM section.
Line 206-208 is this the only conclusion from comparing the studies? What a bout the impact of PA profile?; do we know the time of the experiments? If yes, what is the role of temperature here?
Figure 3. The 3d presentation could mislead the reader, would be possible to change the Figure to a column bar, where the extent of PA-degradation and percentage of senecio are more visible?
Line 252-254: It seems the steepness of curve in Figure 4 is related the specific-interchange of PAs? See my comments for Line 377-386.
Line 269-275: You can elaborate little bit more here by discussing the impact of dry mater content (table 2) on the gas yield, plus the biomass.
In addition, what about the impact of CA, CP, Cl, WSCH from Table 3, which is never discussed!
Line 356-360. So there is effect of the pH in the media? Or this has not been looked into?
Line 377-386: what about the specific-interchange of PAs/ metabolic reduction of N-oxides, which when one PA for example senecionine n-oxide degrade will form senecionine? Therefore, sometime a trend of increasing concentration of one PAs observed sometime.
This is need to be clarify or maybe discussed. Example of references are:
https://doi.org/10.1016/S0378-4274(03)00293-5
https://doi.org/10.3390/molecules24030498
Where is conclusion section? I believe there are other a lot nice conclusions could be made from this study….
Author Response
I would be better to discuss/present the result in the order of the aims. That were discussed in line 94-104. In the current version, the flow is not the same as presented.
Response: We consider the aims as a list of questions that we liked to address, however it is not always possible or necessary that these aims are the guideline to present the results or the discussion, since the topics are very much interwoven and connected.
In addition, the link between sub-sections in RD section is missing. Even though the sub-sections are nicely discussed, however, the are not sell connect with each other. For example, what we will lean from section I?, and what is the additional outcomes from section II? Then are section I and II connected? Or how do they complete each other. Thus, this kind linkage is necessary!
Response: We have taken this into account and modified the paragraph synthesis (chapter 3.) accordingly to facilitate comprehensibility.
Degradation rate and half-life either presented nor discussed, for example in section ‘’2.3.1. Results of the ensiling experiment’’
Response: The degradation rate is presented in Figure 3. Unfortunately, half-life degradation levels cannot be reported because the experimental set-up did not include temporal staging to model the degradation kinetics.
Inter-specific changes / metabolic reduction of N-oxides for PAs are not disused.
Response: In our case, we applied a sum parameter approach to quantify the PAs/PANOs. This comes with the disadvantage that we lose the structural information of the original PAs/PANOs, however we gain the advantage that we comprehensively cover also for possible degradation PA/PANO-metabolites. From all that is known from literature, in our case, all PAs were members of just one type (senecionine-type), and no otonecine-type PAs (or other types) were present. Hence, we do not expect unusual degradation behavior and all PAs should be degraded at a similar rate. The method and the corresponding literature of the PA-degradation during ensiling is already included in the discussion section (3.2) and the relevant literature is cited.
The unit / abbreviation is not mention before, consider define the abbreviation when mentioned for the first time…. Please check throughout the manuscript to define the abbreviations…….
Response: We added the missing information.
Line 94-104, the aim or hypothesis of the study is clear, however the approach is missing ….
Response: We added a short statement but the relevant information is already part of the Material and Methods section.
Lake of scientific literature for section 3.4. Synthesis
Response: We added multiple changes and made several adjustments to the Synthesis paragraph.
PA concentration/content; is not clear the data discussed in result section is generated by HPLC or GC, this need t be clear through out the manuscript.
Response: All quantifications were done by HPLC-ESI-MS/MS sum parameter method, the PA-patterns (this is not quantitative but qualitative) was done by GC-MS. This is very clearly pointed out in the Material and Methods section and of course applies in general throughout the manuscript. In addition, it is stated in the Results and Discussion section or at Fig. 1 as well.
Table 1. You need to define any acronyms or term that used; Dry matter content at start, and Water-soluble carbohydrates with a footnote……and (sd) is from n=? measurements
Response: We now also use the full wording with the respective acronym when it is first mentioned in the text. The number of replicates (n) has been added in the table.
Line 140: Did you define ‘’ WSCH:XP-ratio’’ before, or is it a typo error of WSCH:CP-ratio
Response: We have corrected the error accordingly.
And ‘’ moderate ensilability’’ is rather descriptive, can you define the moderate with maybe scientific figures.
Response: We have addressed the issue: “ …moderate, i.e., sufficient carbohydrate supply for the lactobacilli is counteracted by a notable buffering capacity of the substrate.”
Table 2. Define the numbers is brackets for BBCH, is it number of days? replicate?
Response: The numbers in brackets stand for defined ontogenetic stages of plant development. The first number indicates the general stage of physiological development, the second number is used for fine scaling. The international BBCH-scale is commonly used, and we have cited the corresponding source. So we would suggest to keep it as such.
For ‘’effect of the stage’’ compered with what? Its is not clear!
Response: In this column, the statistic results of the effects of the factor “stage” on the measured variables in the corresponding rows in the table (plant height and DM content) are addressed. This type of tabular presentation of ANOVA is common and we would like to keep it.
Line 154-158: how constituent characteristics correlated with the plant growth? Or for example do we know why the ‘’crude ass’’ decreased with growing season, the same for rest of the constituents
Response: Here, we presented the effects of the stage on the biochemical characteristics, not the correlations since this was not the part of the experimental question. However, some assumptions can be made: Young plants have higher crude ash contents, as it already absorbs many minerals in spring with good water availability. With the onset of elongation growth, there is a dilution effect due to the massive increase in organic components. This reduces the percentage (but not the absolute amount) of minerals in the plant, and so the CA content decreases.
All parameters concerning the structural/scaffolding substances (ADF, NDF, and other cell-wall-related characteristics like EULOS) must increase with increasing plant age and height, as the plant must biophysically stabilizes when its height increases. Since these connections are generally known and were not the focus of our question, they were not discussed in detail and, moreover, we believe that this would go beyond the scope of the discussion.
Table 3, define the units. Was the correlation for ‘’ effect of the stage’’ was positive for all constituents.
Response: The units are given in brackets in column 1. We do not present correlations here, but the test results of analysis of variances (ANOVA and the non-parametric adequate Kruskal-Wallis). These results are neither positive nor negative, they only indicate the significance of an effect irrespective of its direction. With the exception of the crude fat content, however, the trends in the development of the characteristics with increasing ragwort maturity can be obtained by the mean values.
Line 169-12: check the grammar! And some other places…
Response: We addressed the issue.
Line 184-189: the sentences are hard to follow, please consider rewarding them.
Response: We have rephrased the sentence.
Line 195-205: The approach has not been discussed in MM section.
Response: Now, the information was transferred to the Material and Methods section as suggested.
Line 206-208 is this the only conclusion from comparing the studies? What a bout the impact of PA profile?; do we know the time of the experiments? If yes, what is the role of temperature here?
Response: Also, these are not our own results, we think this paragraph belongs to the results section, in which neither discussion nor conclusion is intended. As explained in the text, we have only looked at studies with comparable silage duration (90-105 days). Temperature is indeed an interesting factor that could influence PA-degradation. Unfortunately, the authors of the experiments cited here do not provide complete information on this.
Figure 3. The 3d presentation could mislead the reader, would be possible to change the Figure to a column bar, where the extent of PA-degradation and percentage of senecio are more visible?
Response: Similar problems would arise with a column chart. There is the problem that lower values in the rear area are covered by larger columns in the front, which is why we decided to go for a dot chart. However, we added the raw data for this plot as a table to the supplement. This ensures that there can be no more misunderstandings and the possibility of a quick visual classification remains.
Line 252-254: It seems the steepness of curve in Figure 4 is related the specific-interchange of PAs? See my comments for Line 377-386.
Response: Figure 4 does not contain a curve. If Figure 5 is meant: This impression could arise, which we also pointed out in the relevant part of the discussion (line 413-415 in the original manuscript). With reference to a previous study, however, we refute this putative influence (lines 415-419 in the original manuscript).
Line 269-275: You can elaborate little bit more here by discussing the impact of dry mater content (table 2) on the gas yield, plus the biomass.
Response: This is the Results section, hence discussing the result would be not appropriate here. However, the calculated and presented specific energy potential per unit of volatile matter has already been corrected for the DM effects and is presented in formulas 1-3 in the Methods section. In practical wet fermentation biogas plants, it is the basic liquid substrate (usually liquid manure) and only to a small extent the dry matter content of the co-ferment that determines the dry matter content in the fermenter. Therefore, we have not discussed this subordinate aspect here.
In addition, what about the impact of CA, CP, Cl, WSCH from Table 3, which is never discussed!
Response: Again, in the Results section we only point out the results and the impact of the plant characteristics is described in detail in chapter 3.1.
Line 356-360. So there is effect of the pH in the media? Or this has not been looked into?
Response: Lines 356-360 refer only to the source in question, which unfortunately does not allow to draw conclusions on a general pH effect. The other silage trials cited (i.e., those without alkaline additives), on the other hand, hardly varied in pH value, so no tendencies can be derived from them either. However, this is an interesting point to look at, but needs to be addressed a separate study.
Line 377-386: what about the specific-interchange of PAs/ metabolic reduction of N-oxides, which when one PA for example senecionine n-oxide degrade will form senecionine? Therefore, sometime a trend of increasing concentration of one PAs observed sometime.
This is need to be clarify or maybe discussed. Example of references are:
https://doi.org/10.1016/S0378-4274(03)00293-5
https://doi.org/10.3390/molecules24030498
Response: Yes, we are aware of a possible reduction of PANOs to tertiary PAs in plant degradation (however interchange, per definition in both directions seems impossible/was never observed in plant degradation). As discussed above, for quantification we used a sum parameter approach, covering for all 1,2-unsaturated ester PA/PANO including unknown metabolites. Since both forms of PAs (tertiary and PANOs) are considered to be toxic, the main question (possible downstream toxicity) is very well reflected with our approach, since it also covers for possible structural related but unknown PA/PANO-metabolites which would be missed in a target-analysis approach covering only the know (and covered by available reference compounds) pairs of PAs/PANOs. In addition, so far in plant degradation it is very well known, that PANOs are to some extend converted to tertiary PAs, but overall the sum of both decreases by a large amount. Hence, this overall degradation is very well covered with our approach since we always analyze the sum of both, at the beginning and at the end, and we can nicely confirm the overall degradation, which is/was our main goal.
Where is conclusion section? I believe there are other a lot nice conclusions could be made from this study…
Response: We have addressed the issue and included a new chapter Conclusions that goes beyond the Synthesis of the first version of this manuscript.
Reviewer 3 Report
Overall impressions
The manuscript is well structured and written. The experimental section is well planned and consistent with the results. Although the authors present a properly and extensively elaborated discussion, the conclusions are lacking. Consisting-answers to the five research questions stated in the Introduction must shortly be written in the Conclusion.
Minor comments
Figure 1: Quality is not acceptable for a publication.
Line 551: Must be H2SO4
Author Response
Consisting-answers to the five research questions stated in the Introduction must shortly be written in the Conclusion.
Response: We have addressed the issue and summarized the concluding passages from the individual chapters of the discussion in a new chapter "Conclusions".
Figure 1: Quality is not acceptable for a publication.
Response: The image has been revised.
Line 551: Must be H2SO4
Response: We corrected the mistake.
Reviewer 4 Report
Dear Authors,
The title of the article sounds interesting. Also, performed analyses look interesting.
The article does not require significant corrections.
The introduction is comprehensive and introduces well the research topic,
In my opinion this work should make a good contribution to the literature. Further more, in my opinion, the paper will attract a wide readership. The MS looks good. Therefore, I have no serious substantive comments to the MS.
The experimentum design and data analysis are appropriate, the introduction and discussion are consistent.
Recommendations for authors:
However, the authors need to correct the technical shortcomings:
1. Figure 2, 4 disproportionate to the rest of the text,
2. Depending on the chapter, some lines aligned some not eg. chapter 4- Materials and Methods
3. Please read the MS once more and correct any minor shortcomings, e.g. punctuation, etc.
Author Response
Figure 2, 4 disproportionate to the rest of the text
Response: Both illustrations are vector graphics and can therefore be changed in dimension without loss of quality. The current proportions were chosen so that the illustrations fit well into the block text and unnecessary empty spaces are avoided. In the case of a contribution acceptance, we can comply with the wishes of the layout team easily.
Depending on the chapter, some lines aligned some not eg. chapter 4- Materials and Methods
Response: We have aligned all chapters accordingly.
Please read the MS once more and correct any minor shortcomings, e.g. punctuation, etc.
Response: We have addressed this issue.
Round 2
Reviewer 2 Report
The revised manuscript has improved compared to the original version. The Authors tired to address my question as much as possible.
Overall, the quality of the written text with respect to English phrasing and grammar is good and acceptable.
Best wishes,